# Between an Acknowledgment of Immigration and Neglect? Assessing Interculturalism and Media Integration in Luxembourg

Suzana Cascao

Contemporary History of Luxembourg, Luxembourg Centre for Contemporary and Digital History (C2DH), University of Luxembourg, Esch-Belval, 4365 Esch-sur-Alzette, Luxembourg; suzana.cascao@uni.lu

**Abstract:** Luxembourg is a de facto multicultural country, with 179 different nationalities represented. It is, however, complex to identify who among the latter is perceived as an immigrant by public opinion. In the same vein, immigration stories rarely make the headlines of some of the most prominent outlets of Luxembourg's mainstream media. This study covers the print content of some of Luxembourg's dominant media outlets in the search for the representation of (im)migrants and refugees. It thus takes a perspective whereby media act as a vehicle for a quintessential aspect of interculturalism, that of local meaningful interaction. Its overarching question regards the role that both local mainstream and minority media sectors can play in promoting integration through intercultural dialogue. It is hereby argued that immigrants are foremost represented and given a voice in media outlets created for the immigrant and cross-border communities as well as in mainstream media with more local (*Tageblatt*) and independent political views (*D'Lëtzebuerger* Land). In stronghold media such as *RTL Lëtzebuerg* and *Luxemburger Wort*, immigrants are, instead, scarcely represented.

**Keywords:** intercultural integration; multiculturalism; mainstream media; minority media; Luxembourg; migrants

## 1. Introduction

Interculturalism requires places for interaction and dialogue to happen. This empirical study links up with bigger pressing questions in the current intellectual debate on immigration and intercultural coexistence. In doing so, it takes media and the quest for intercultural integration as essential. Taking as a point of departure the notion of intercultural dialogue, this research discusses how it can apply in the media. What often fails to be identified as a key aspect of interculturalism is how it can be rendered visible and practiced also in terms of integration through media presence, and this is what this paper is concerned with. Existing research is indeed still incipient on the matter of interculturalism applied to the media. Its overarching question is that of the role that both mainstream (majority) and minority media sectors can play in promoting integration through intercultural dialogue. In this vein, this research is based on the premise of the inextricable relationship between the media's power to forge public opinion and the phenomenon of immigration in culturally diverse societies. The basic elements of the concept of integration are the idea that cultural heterogeneity must be framed by a recognition of commonly accepted and respected fundamental values and that the search for the optimum between homogeneity and heterogeneity is at the core of the problem of integration (Geißler 2015). Furthermore, and relevant for the research case in point, integration is here understood as simultaneously a scientific–analytical and a normative–political concept (Schnapper 2007) Through her societal approach, the Schnaper (ibid.) concludes that over time, the political meaning of integration has become divorced from its strictly sociological meaning, leading to a trivialisation of integration as an issue that only concerns the behaviour of specific communities. By ascribing to the

work place its role as one of the main arenas of integration, the author highlights the offset between societal norms and the real possibilities of the disenfranchised to conform to them. Intercultural dialogue operates optimally at a local level, by way of interaction between different communities. At the basis of the present research's understanding of the potential role of media is, therefore, the application of the concept of interculturalism at a micro level, where individuals and local institutions are delegated with the task of managing and negotiating diversity. For this reason, the relationship between immigration, media, and the local reality—that of a small country specifically—seems pertinent.

Assuming that intercultural dialogue is optimal and desirable, in which capacity can content conveyed by media result in greater interaction between immigrants and the host society or simply allow for a more objective knowledge of one another? Is there space in the Luxembourgish mainstream media for a storytelling of immigration that concedes both visibility and an unbiased account of immigrants, refugees, and minorities' daily lives?

As is known, multiculturalism in Europe and elsewhere has faced over two decades of backlash. The severe criticism placed on multiculturalism ever since has been well documented by a number of social scientists (Macey 2012; Brahm Levey 2012; Modood 2013; Johansson 2022) and needs little more than a brief incursion here.

This paper offers a reflection from the standpoint of interculturalism; however, in the attempt to understand what interculturalism has to offer as a novelty, also in terms of mediated communication, it needs to acknowledge where its criticisms come from. That criticism is often posed in terms of what is it that intercultural policies, practices, and paradigms altogether offer that differs from its predecessor, multiculturalism (Barn 2012; Mansouri and Modood 2020).

More recently, the emergence of concepts such as conviviality (Gilroy 2004) and superdiversity (Vertovec 2022) have also contributed to the debate.

Multiculturalism has seen many strands of theoretical assessments, political and policy models. It is a term so fluid and contested that it led to "confusion and conflation between the academic and theoretical debate about the concept on the one hand and the way the concept is used by the media and public opinion on the other" (Johansson 2022, p. 3).

Furthermore, the debate on multiculturalism both academically and politically has seen several top politicians declaring the failure of multiculturalism (Barrett 2023) Yet, multiculturalism is still being blamed, even more than two decades after its death has been declared (Kundnani 2002). Among its fierce critics, Suella Bravemann (Home Secretary of the United Kingdom) said immigrants to UK should integrate because "multiculturalism as end in itself' could lead to disaster" (Malik 2023). The widely accepted paradigm in favour of diversity is now being offered by interculturalism. Interculturalism emerges as an academic critique to multiculturalism whilst claiming a pro-diversity stance, which is presumably also a more pragmatic policy perspective (Mansouri and Modood 2020). As new research approach it consists primarily not in theoretical but in "empiric, often local or policy-oriented analysis" (Mansouri and Modood 2020, p. 3).

This novel research approach has permeated into the governance level and is now widely accepted in Western Europe. Another strand of interculturalism, highly relevant for the Luxembourgish case because of context similarities, is that of Quebequian interculturalism (Bouchard 2013; Symposium International sur l'Interculturalisme 2023) but we will not pursue this here. This research is interested in seeing how interculturalism, in its pragmatic approach, officially endorsed by the Luxembourgish government, applies in terms of the messages conveyed by the country's media actors. For this reason, the relationship between immigration, media and the local reality appear as an ideal arena in which to put interculturalism at test.

Interculturalism took the scene in Europe, endorsed by the Council of Europe and through its White Paper on Intercultural Dialogue, in 2008, but it is only 15 years later that the tide of interculturalism will translate into an official policy in the Grand Duchy. The current Grand-Ducal law in place on integration,[1] since 2008, which foresaw at a certain stage an "integration contract", faced some criticism. Its definition of integration had some

arguing that it supposed an "individualist approach (...) and one expressing wishes of voluntary capacity from the immigrant only" (Serré 2010). The point worth noting still according to Mansouri and Modood (2020) is that:

> while most European governments have moved in the direction of an interculturalist paradigm focused on the local governance it is interesting that their anti-multiculturalist rhetoric has not been substituted by interculturalism, disregarding their being signatories to the Council of Europe's promotion of interculturalism". (Mansouri and Modood 2020, p. 7).

It is worth highlighting that interculturalism has been increasingly gaining ground even among those scholars (Mansouri and Modood 2020) who have remained the last bastions defending multiculturalism. As interculturalism becomes mainstream in the political arena, it is thus useful to emphasise here where some of its originality and strengths come from:

(a)  In its focus on a local dimension, moving from a state-centered approach to a local-centered approach in diversity policies (Zapata-Barrero 2015);
(b)  In its stance towards immigration as a resource;
(c)  In its meaningful (positive) interaction aspect;
(d)  In its fundamental element of equality and access to citizenship.

Intercultural dialogue, therefore, entails a bottom-up shared responsibility that is no longer only that pertaining to the state but also to the host society at large (Wilson 2013). Other authors claim: "It was about a gentle but firm obligation placed on all of us to engage in an active negotiation of meaning across assumed cultural boundaries" (Sondhi 2008). Thus, the public and private sectors (including the business world), pressure groups, individuals, and the media are all involved in putting intercultural dialogue to work. Interculturalism requires a forum, a space for interaction and dialogue to take place. Interculturalism further highlights the role of education, municipalities, and individuals in a bottom-up approach that nevertheless lacks, still, "any clear theoretical framework to legitimate their new policy approach (Zapata-Barrero 2015, p. viii). Specifically, what comes into question, in the framework of this brief study, is the inextricable relationship between immigration and the media and the ways in which intercultural interaction may be mainstreamed.

Within the decades-long debate opposing interculturalism to multiculturalism, different features and contexts of diversity in different geographical settings[2] need to be taken into consideration. One of them regards a multilingual (trilingual in Luxembourg) situation that precedes, by centuries, that of the first significant immigration waves into the country, whereas the other is that of a "threatened "majority", with Luxembourgers making up just over the majority at 53%.[3] It could be further mentioned that in many localities, immigrants (specifically some nationalities) represent de facto minority–majorities,[4] with the capital of the Grand Duchy Luxembourg City being a case in point. Furthermore, a significant number of Luxembourgers are of foreign descent. The jus sanguinis prevails over jus solis in Luxembourgish jurisdiction, making it so that children born and raised in Luxembourg to foreign parents can only assess nationality by the time they reach 18 years of age.

In regards to access to citizenship for foreign adults, significant progress took place in Luxembourg over the last few decades. The rights for foreigners to vote in municipal elections has been widely facilitated, whereas in the case of a vote in national elections, it has been rejected by referendum in 2015. However, the participation of foreigners is far from being ideal and is still surrounded by debate, particularly in 2023, a *Superwahljoer* year in which both municipal and national elections take place. In fact, only one in five foreigners were expected to turn out to vote, the reasons behind this apparent disinterest being varied, for example due to language-related issues as hinted by some media outlets (Letzebuerger Journal 2023).

Predictably, the participation of foreigners is presently high on the media agenda and represents a significant amount of the news items dedicated to immigration.

Almost half of Luxembourg's resident population is composed of foreigners. With over a century-long tradition of immigration, mostly European, the origins of its foreign population have been progressively diversifying to include countries outside Europe. The majority of its foreign population is nowadays composed of Portuguese (14.8%), followed by French (7.6%), Italian (3.7%), Belgian, and German nationalities. Added to the (resident) presence of French, German, and Belgians is the number of 200,000 daily commuting foreigners from the latter three neighboring countries, which needs to be added to the count of diversity. This often falls into a grey zone in terms of its conceptualisation and dynamics in terms of integration politics. The OECD integration country report of 2021 (Breem et al. 2021), not only reflects the many nuances of the origins and typology of foreign presence as it also highlights the significant number of youngsters with a migrant background. It further underlines that a quarter of the immigration arrivals stem from other countries such as China, Brazil, and India (Breem et al. 2021).

Very peculiar to Luxembourg is its three official language system (Luxembourgish, French, and German), whose use varies according to the context, rather than following a regional difference unlike what happens in other European countries. Any of the three languages can be deemed necessary (vide mandatory/essential) within different stages of the integration process. Indeed, the complex language and educational system has been identified as one of the reasons for high rates of poor schooling results (Horner and Weber 2008). This is particularly impactful among youngsters with a migratory background as the different languages are introduced, gradually, at different schooling stages[5].

This situation has recently led to a call to action from the EU, which warned against the consequences of such a system, specifically among children of a migratory background and of a disadvantaged socio-economic milieux.[6] Different sociolinguistic research has also focused on this matter (Heinz and Fehlen 2016a; Vasco Correia 2013). A plea for change and flexibility in the choice of a vehicular literacy language has been widely debated and endorsed by a part of the political establishment as well.[7] Horner and Weber (2008) also highlight the added complexity when such migrants, youngsters from a migrant background, or binationals already use different or more than one language at home. In their study of families of Greek immigrant background in Luxembourg, Kirsch and Gogonas (2018) further demonstrated the inextricable relationship between language ideologies and language policy as well as the strain exerted by the three languages of instruction system at school. The current system exposes children of ethnic minority backgrounds to considerable difficulties. The latter research is an example, furthermore, of the need for studies considering groups that have one or more (other than the three official) languages spoken at home. This is the case of an ever-increasing number of children of dual nationality in Luxembourg.

German thus remains the literacy language of Luxembourgers as well as the language vehicle in most of Luxembourg's traditional mainstream print media. Daily communication, as well as the language vehiculated in radio and TV, is Luxembourgish (official since 1984), which creates the paradox of German being simultaneously a compulsory language in the school system but a de facto "dead" language in the day-to-day spoken interactions among Luxembourgers. The language circumstances in such a small country inevitably lead to adaptations in communication according to context and audiences. For example, English and Portuguese are widely used (in addition to French and Luxembourgish) in local administration communications, to the detriment of German which is increasingly in disuse in municipal communications and political and other type of campaigns aimed at the public. In the job market, research shows French is a must (Heinz and Fehlen 2016b).The role of languages in shaping Luxembourgish identity should thus be understood within complex and changing political, ideological, historic and economic frameworks. The idea of what means to be a Luxembourger started being shaped namely at school and through the changing narratives of the country's most prominent historians (Peporté et al. 2010; Rohstock and Lenz 2011). Language conflates with nationality and nationality with identity, although they should not, according to Trausch, be mutually exclusive:

> However, it would be inappropriate to make Luxembourgish the sole criterion for Luxembourgish nationality (...) A nation is not based solely on a common language (...) it is based first and foremost on the desire (...) to live together. (...) There can be no nation without solidarity; there can be no nation without the recognition of a higher general interest (Trausch 1985).

The mastering of German and Luxembourgish is indeed what differentiates old Luxembourgers from new Luxembourgers, many of whom hold nationality without really mastering either German or Luxembourgish (despite the latter being a prerequisite for nationality). Here, languages come into a play as a central and foremost factor of integration (Christophory 1978), as in almost every situation of day-to-say life, not knowing one of the official languages may preclude one from feeling they belong[8]. In the same vein of Bourdieu's (Bourdieu 1982) understanding of language as mechanism of power, Luxembourgish acts as the ultimate requirement for integration; paradoxically, this was a dialect turned into language, much the opposite of the examples in Bourdieu's aforementioned work. In fact, any of the three official languages can be waived as a decisive criterion in fronts that range from the education system to the job market, to political campaigns. The printed media is rich in heated debates about the place of each of the languages (Kollwelter 2023; Thill 2023).[9]

The unease in the use of one or more languages can translate into the job market, where, although French is the lingua franca among the lower-qualified jobs (Heinz and Fehlen 2016b; Fehlen 2013), Luxembourgish is increasingly in demand. At school, on the contrary, Luxembourgish acts as a vehicle of integration, but German does not[10].

It is therefore only natural that the media landscape follows suit to adapt to each of its ever-growing new audiences.

On both the mainstream and minority[11] media fronts, the mediated/communication context assumes the utmost relevance. Many of the challenges arising from academic research privileging an interculturalist prism are associated with the difficulty of putting the latter into practice. The media system has been identified as one potential vehicle for an intercultural type of integration.

> Information should be made available at various levels and in all spheres of society about what is required of members of the majority and the minorities. Likewise, public debates should be promoted in numerous venues with the collaboration of the national and local media. (...) Still in the cultural sphere, immigrants and members of minorities should be made more visible in the media and public institutions so that they become part of the cultural landscape. Diversity should be displayed everywhere in order to break the old, embedded vision of a homogenous society, to prevent the formation of boundaries and help people come to terms with the new ethno-cultural order (Bouchard 2013, pp. 106–7).

These spaces for intercultural dialogue are to be addressed as complementary components of a highly participative plural democracy. Such participation, as seen, can be attained through public support for immigration (Beck 2003; Geißler and Weber-Menges 2009; Zapata-Barrero 2015), faster access to citizenship,[12] and a reduction of prejudices, all of which can be initiated and promoted by the media.

The mainstream media's modus operandi in collecting and conveying information about migrants and minorities has been widely covered in research, but the relevance of the main actors in the phenomenon remains to be sufficiently explored: migrants themselves (Bleich et al. 2015). The latest European-wide media debates around immigration had been increasingly centred on boundary defence, securitisation and threats within the nation–state, be it in politics or in the media (Caviedes 2015; Triandafyllidou and Ulasiuk 2015). There is a large number of content analyses that show that many media—especially large sections of the print media—convey a negatively tainted and distorted image of ethnic minorities (Weber-Menges 2015). However, recent evolutions within some media outlets meet the changes in journalism practices highlighted by some publications "which have found, instead an "increased investigative reporting, counter-argumentation and use of

various sources of information as indicators of less essentializing forms of portrayal in the news"(Triandafyllidou and Ulasiuk 2015, p. 156).

The most recent contributions in the field of intercultural integration and media (Triandafyllidou and Ulasiuk 2015) highlight the methods and a paradigm of intercultural media integration that follows the lines of a model proposed earlier by German scholars Geißler and Weber-Menges (2009), proving its pertinence remains. The latter describe the integrative model of media as one in which the local host population and minorities mingle, thus allowing for intercultural communication to take place. Homogeneity is not expected, but rather mutual knowledge and communication about the differences between the communities (host and newcomer). Intercultural media integration supposes a proportional participation of minorities in the production of majority/mainstream media (Triandafyllidou and Ulasiuk 2015) as well as formulating the ideal functions of targeting minorities' media, by expecting that migrants with knowledge of the host society produce such media in a way that promotes intercultural integration. At the same time, mainstream media is urged to echo minorities more frequently and more visibly while promoting adequate and balanced coverage of the issues that immigrant communities face. Moreover, the characteristics of interculturally inclusive media content can also be formulated negatively: for example, ethnocentric media that do not give minorities enough space to express themselves and do not consider their sensitivities and problems. However, this does not mean that problems with migration in the host society are to be treated as taboo. They are, just like gendered or generational problems, part of the pluralistic open discourse (Geißler 2015).

Minorities naturally search for a space of (counter) representation. Research on minority media has pointed out that "minorities also represent themselves (and others) in their own media, those that they consume in and across the societies in which they are minorities, as well as those that they consume in such (dis)locations" (Silverstone and Georgiou 2005, p. 434).

Several authors consider that minority media can support and act as a means of integration by giving visibility and participation to minorities in the general public sphere (Georgiou 2006; Matsaganis et al. 2011). Nevertheless, other studies on media pertaining to minorities (Becker 1998; Becker and Behnisch 2001) point out that its use, especially if prolonged overtime, promotes immigrants' own "ghettoisation" (Kim 2001) considering the engagement with minority media only as an exercise of hermetic consolidation of their original cultural identity. Conversely, the academic tradition in the field of media has increasingly coincided with the advocacy for cultural pluralism whilst it defended the maintenance of some degree of "ethnicity" (Kim 2001). Minority media undoubtedly back the specific needs of the immigrant/minority communities that they serve as a representative. They fill the space left empty by the national mainstream media and provide migrants with a forum for self-representation (Kosnick 2007).

The balance to strike is ever-growing in complexity in highly diverse societies. When minority media initiatives can act independently, outside the frames of a bigger national editorial group or broadcaster, they will "celebrate the subjective aspects of immigrants, strengthen the dynamic of interethnic/intercultural recognition and calls for maintaining the ties with one's own culture, country of origin and people" (Gabellieri 2002). This situation will, however, have another side to it, as Gabellieri (2002) argues further: "that of an estrangement from the editorial context, and potentially from the social fabric of the host country" (Gabellieri 2002, p. 12). In it lie the main arguments against supporting the existence of independent minority media, in that they pose the risk of immigrant and minority communities segregating themselves, thus posing a threat to meaningful interaction. In terms of what the mainstream media conveys, instead, in their message about immigration, we saw that what interculturalism preconises as desirable includes a positive outlook on migration and the inclusion of minorities, namely, in the production of media content.

The offer in the media market in Luxembourg is large but "highly fragmented in linguistic and cultural terms"(Kies et al. 2022). The positions of the editorial groups RTL and Mediahuis in the market are uncontestably dominant. RTL recently had to extend its public mission by fulfilling new obligations such as: "the promotion of media education and of the local cultural scene" (Kies et al. 2022, p. 14). Issues such as social inclusion and participation of minorities in the media are also pressing according to the same Media Report. It is unfeasible to determine the nationality of journalists as there are no statistics available. The results are nevertheless clear from the analysis that the journalists working for the minority media, in Luxembourg, have privileged access to the communities they write for and of their respective language. It can thus be said that there is the participation of minorities, or of journalists of a migrant background in the minority media on offer, selected for this study. More complex to identify is whether there are journalists representing minorities in the mainstream media, but as we shall see, there are other ways to translate and transfer content to and from one another (majority mainstream media to minority media and vice-versa).

In the same vein, other key aspects of the assessment of intercultural media integration, such as the participation of minorities in the production of media, are also here—albeit superficially—covered. Because of their quintessential role in determining the traits of intercultural media integration, it is therefore acknowledged that the dynamics of minority participation in media production, as well as the role of audiences, deserves further and urgent attention from in future research, because they are imperative for understanding ways of relating to the media offer in different languages. Furthermore, the lack of studies on the impact of the media representation of ethnic minorities in the Luxembourgish media is also because empirical research on this topic itself faces several difficulties, namely those of finding researchers who master all three official languages of the country, plus eventually others. As happens in other parts of Europe, the concrete contents of the minority media remain "in the dark" (…); linguistically skilled observers and researchers are needed to "bring them into the light" in the host society" (Geißler 2015, p. 72).

## 2. Methods

This study reflects on print content and where and how intercultural interaction is conveyed in the two media sectors (mainstream aimed at host society and minority media, respectively). Using the model of intercultural media integration model as a blueprint (Geißler and Weber-Menges 2009; Geißler 2015) we analyse how the concept responds when operationalised in the field of communication. In the integrative, intercultural model pre-conised by Geißler and Weber-Menges (2009), majority and minorities are "intermeshed". Not only does the model advocate for a participation of minorities in the mainstream media, as it entails that migrants with knowledge of host society produce media themselves, another aspect of the intercultural integrative media model is that it transposes the very principles of interculturalism into its contents, namely the explicit acceptance and "necessity" of immigration. These aspects are to be transposed to the mainstream (majority) and minority media dynamics of production, representation, participation of migrants, refugees, or minorities in the making of media content. Finally, the model makes a case for the use of both minority and mainstream media from those who are able to master more than one language. Regarding research about aspects of consumption and audiences of such media, Luxembourg is still lagging behind in terms of data available (Kies et al. 2022).

By contrasting communication coverage across both mainstream (Luxembourgish- and German-speaking) and minority outlets—in this instance of Portuguese-, French- and English-speaking minorities in Luxembourg—this brief study offers a unique and original analysis of media stances on intercultural dialogue. It can be said that in Luxembourg, being a very small country, the national mainstream media supplant the existence of a regional press (with the exception of *Tageblatt*, a newspaper founded for and still aimed primarily at the audiences of Luxembourg's second largest city, Esch-sur-Alzette). Therefore, national mainstream media overlaps as regional and local media as well.

The methods chosen privileged an observation of media production, whether it be mainstream or minority media. As a principal indicator, we privileged content assessment (Supplementary Materials). Media content, featuring diversity in the shape of immigration, refugees, and minorities, can be approached in a myriad of ways: in-depth, minimal, positive, negative, folkloristic, social, economic, or political.

This study did not use any text processing analysis. It was also less concerned with a quantitative rather than with a qualitative analysis. The analysis required a close reading of textual matter (articles), which, although not entirely avoiding quantification, privileged content featuring some categories (topics depicting immigration and or refugees) and its attached narratives and typology (informative, cultural/societal, historical, opinion, and political). During the analysis period, no further typologies emerged. More importantly, we looked for the occurrences in which the content focused on the immigrant as the source and agent of their own narrative or where topics covering their sensitivities were showcased. Moreover, we searched for occasions where immigration was indeed portrayed as accepted and welcomed as a necessity.

The selection of mainstream majority media was rather straightforward and based on circulation numbers (Javel 2023) as well as on an attempt to portray media catering for a Luxembourgish audience but of mixed "sympathies". This research thus included print media that, on the one hand, represented media in circulation having the host society as a target audience:

(1)    The group RTL, which is also vested with a state mission (Kies et al. 2022), in this instance particularly its news content web page *RTL Lëtzebuerg*;

(2)    The newspaper *Luxemburger Wort*[13] (belonging to the Mediahuis editorial group, audience of 136,800 (TNS ILRES 2020));

(3)    *Tageblatt*,[14] (part of Editpress), which allowed us to include an historical newspaper aimed, primarily, although not exclusively, at a local audience: Luxembourg's second largest city Esch-sur-Alzette, where 120 different nationalities live side by side.

(4)    Finally, we chose to include *De Lëtzebuerger Land*, an independent newspaper[15] that, despite having smaller circulation numbers (16,900[16]), deserved our attention because of its positioning with regards to Luxembourgish society and politics when trying to understand the different editorial stances in the discourse regarding migrants and refugees.

Unavoidably, some print media with significant circulation numbers were left out of the selection,[17] primarily because of the difficulty in categorising them into mainstream or minority media.

One factor deserves attention in the identification of the subtle differences between majority and minority media, and that is that even within the German-/Luxembourgish-language media, content in French[18] is also available, within the same newspaper edition, albeit admittedly in smaller numbers with respect to German-language content.

As for the rationale behind the selection of minority media, the circumstances in the Luxembourgish media landscape increased the complexity of the exercise and, namely, who is to be considered a "numerous" minority, who manages to produce media in a significant scale and consequently translates it into circulation numbers/audiences. The selection, therefore, had to be composed of a steady and consistent presence in the market, as well as in languages available for the researcher to understand. This aspect should not go underrated, as minority media content is often barred from the researcher because of their inability to understand other languages (Matsaganis et al. 2011).

The boundaries here lie with the difficulty in identifying independent minority media outlets in Luxembourg. Indeed, the totality of minority media here under scrutiny belong to the same two groups owning a significant portion of the mainstream majority media outlets, RTL group[19] (*RTL Lëtzebuerg*, *RTL Today*, and *RTL Infos*) and Mediahuis (which holds Contacto, Luxemburger Wort, the Luxembourg Times and Virgule), see Table 1. This situation is echoed by the Monitoring Media Report for Luxembourg, as a critical area.

**Table 1.** Split of media selected for analysis, with mention of circulation numbers and publishing groups.

| Minority Media | Mainstream Majority Media |
|---|---|
| *Contacto*—daily digital content in Portuguese language, Mediahuis groupAudience: 47,000[20] | *Luxemburger Wort*—Daily newspaper with a circulation of 148,300 (paper and digital versions included), in German language, Mediahuis group |
| *Virgule*—daily digital content in French language, Mediahuis group[21] | *RTL Lëtzebuerg*—daily digital content in Luxembourgish, |
| *Luxembourg Times*—daily digital content in English belonging to Mediahuis group[22] | *D'Letzebuerger Land*—weekly newspaper with contributions in (mostly) German, French and occasionally in English, circulation of 16,900[23] (independent publisher) |
| *RTL Today*—daily digital content in English | *Tageblatt*—daily local newspaper mostly in German with contributions in French, circulation of 32,000 (Group editpress) |
| *RTL Infos*—daily digital content in French | |

The audiovisual and online sector is dominated by the RTL Group and the printed press is dominated by the groups Mediahuis and Editpress (Kies et al. 2022, p. 9).

From this state of play, two aspects stand out:

(a)    The difficulty in finding independent minority media[24] (or at least whose circulation/audiences are significant outside the above-mentioned editorial groups);

(b)    The realisation by such editorial groups that a market for minority media exists and is coveted.

This is "good news", proving that a space exists for (at least some) minority audiences. Bearing in mind interculturalism's reciprocal approach, where the interaction between immigrant minorities and host society is advocated, this research devoted equal attention to the mainstream (majority) and minority sectors' practices in Luxembourg's media system.

In the case of minority media, it is hereby acknowledged that, for instance, some well-implemented English language media were left out because of their specific content and audiences[25], which were considered/deemed to be less generalist. The same issue presented itself with French-speaking minority media, as virtually all media whose content is exclusively in French are aimed at a cross-border community rather than at immigrant communities who might have French as their lingua franca. An exception to this audience logic seems to be the Portuguese community, who, because they are "automatically" excluded because of the unlikelihood of mastering both German and Luxembourgish[26], end up turning to French-speaking media in the absence of, for example, Portuguese-language content in RTL. As corroborated by one of the journalists at *RTL Infos*, as soon as there is content portraying the community:

> Obviously, the idea of starting this series [*Les Portugais du Luxembourg*, short videos] was prompted by an observation: content on the subject has always generated a lot of clicks on our sites. (...) any content linked to the Portuguese community in any form (portraits, the Wiltz pilgrimage, historical or sociological subjects) systematically makes it into our top 10 most-read articles, and more often than not into the top 5 (Interview with Jerome Didelot, freelance journalist at RTL Infos (Cascao 2023)

The online sites of *Wort, Contacto, Virgule,* and *Luxembourg Times* were browsed once a week on Fridays with focus on their local and cultural sections. The same procedure was used for RTL pages. Content[27] covering immigration and specific immigrant communities were covered. The analysis focused on the local, political, and cultural sections, three

dimensions where immigration was deemed to eventually receive coverage. The data collection took place between mid-January 2023 and mid-July 2023, and also took into consideration the upcoming local elections of June 2023.[28] In total, 211 articles were collected for analysis in the period between mid-January and early June 2023. In addition, the paper versions of *Luxemburger Wort* and *Tageblatt* were scrutinised on their weekend editions. Since *De Lëtzebuerger land* is a weekly newspaper, only the paper version was consulted. Such sample methods do not exclude having missed articles related to immigration that may have been published during the period of analysis.

### 3. Representation, Framing, and Focus of Migrants

The first realisation is that there is space, within the mainstream media landscape, for at least some of the minority communities to have a "voice", and certainly to those whose languages are significantly spoken. What seems to be missing is getting the message across to communities. Host society is, naturally, not required to know the languages of minority communities and thus those of their media. Similarly, content in French, although understood by most Luxembourgers, does not necessarily appeal as a natural language to access media content. As a concrete example, the videos available promoting the lives of the Portuguese immigrant community in *RTL Infos* are not translated into Luxembourgish and do not make it to the Luxembourgish-language webpage (*RTL Lëtzebuerg*) of RTL's website. The editorial decision of conveying a message about an immigrant community beyond the clichés is laudable and admittedly comes in line with an intercultural undertone. However, the content is not translated into Luxembourgish and does not make it to the Luxembourgish site, thus falling short to contribute in making one of the most numerous immigrant communities known to host society audiences.

Another aspect emerges from the RTL information website in its different languages. Whereas Mediahuis group clearly allows for an editorial freedom of choice of content from its minority media counterparts, the RTL information pages in English and French languages are, for the most part, a literal translation of the original in Luxembourgish.

*Content Assessment*

Whereas there are signs of an editorial openness, in this case in Mediahuis, towards different linguistic audiences by creating content that is partially aimed at representing them and their interests through its different minority media outlets (see Table 1), it is unsuccessful performing the reverse; that is, portraying those same communities to the autochthonous population in the German version of the paper, the one a significant portion of Luxembourgers read. The same can be said for the RTL group's media content: when accessing its English and French speaking sites, their take at inclusion is visible mostly in the shape of translations of content they take from the Luxembourgish original, and is thus rather institutional and figure-oriented (foreign populations in numbers, refugees in numbers, representation in the local elections, etc.). It is rather the fact that the occasional content portraying diversity is faint when spreading to an autochthonous audience in the Luxembourgish version of the site. In some of the dominant media (see Table 1), like the *Wort* and *RTL Lëtzebuerg* what reigns is an institutional outlook over the narrative(s) on integration, although there are occasional opinion articles tackling, for instance, the participation of foreigners in the elections, the most recent studies on linguistic and integration measures, etc. The political personalities behind the Integration law receive more of a chance to be in the public eye than civil society's opinions about something that should concern them all. Also, the new Integration law and the coexistence pacts (*Pakt vum Zesummeliewen*) receive an overtly sceptical position around their efficiency from the daily conservative:[29]

> Do you seriously believe that Luxembourgers will sit down with non-Luxembourgers at the weekend in a course in which the country's political system is explained to them? (*Luxemburger Wort*, Interview with Minister Corinne Cahen, (Javel 2023)

*Luxemburger Wort* gives prominence to language-related issues on both fronts: those who defend the maintenance of German as the literacy language and those who increasingly plead for the creation of a school curriculum in French. This took place in the shape of editorials, but also correspondence from the readers, in three occurrences. Overall, content on immigration features mostly as statistical information on the numbers of foreign residents and asylum seekers. Content relies on institutional views of immigration, with incursions on the repercussions of the European asylum policy, the situation of housing for refugees, the upcoming Integration law, and the most recent offer on integration social services. What misses from the narrative is the immigrant as told in the first person, their views, their story, and the immigrant as the central actor or source of the story. The only exception are Ukrainian refugees whose testimonies on their lives in Luxembourg were first covered on the first anniversary of the Russian invasion in *Luxemburger Wort*.

Luxembourg is currently home to a significant number[30] of other asylum seekers and refugees, to whom only *Contacto* and *De Letzebuerger Land* gave a voice during the analysis period at stake, thus going beyond the figure-centred context.

Minority Media

English-language minority media

Minority media in English (see Table 1) respond principally to the needs of a very specific community: the one employed in corporate and financial sector. It fulfills its role of facilitator between immigrant community and host society, by providing all kinds of information on Luxembourg practicalities for the newly arrived, and Luxembourg's history and traditions. It is, however, unsatisfactory in the way it does not represent the different communities who might have English as their lingua franca. Furthermore, the content indeed represents and aims at a community whose jobs are oriented towards financial institutions and international organisations, especially in the case of *Luxembourg Times*. Content featuring immigration overlaps with a self-perception of the community as "expats" working in the finance and service sector. Both *RTL Today* and *Luxembourg Times* display an "expat" section. The personalities who deserve "primetime" are often somehow related to the business world. Minority media in English represent a clear case of selected intercultural representation, with scattered exceptions where, for example, cultural festivities from different cultures around the world are featured. This distinction between "expats" and immigrant presents itself as, admittedly, problematic to the researcher on migration issues. The choice of the term by English-language media is almost self-explanatory of the idea and representation of an "exclusive" English-speaking community, and marginalises other immigrant communities as not being noteworthy of mediatic coverage (an exception exists of two instances highlighting Asian cuisine and restaurants as well as Middle Eastern and Hindu religious celebrations). The concept of "expatriates" has, thus, been widely accepted as "people similar to us", because they are more educated and generally white (Deo 2012; Koutonin 2015). In the case of Luxembourg, this is further unquestionably connoted with being successful and working in finance.

Portuguese-language minority media

The Portuguese and Portuguese-speaking communities stand out as those receiving the most coverage during the period of this study. This is naturally explained by both the frequent content release of the newspaper *Contacto* and a dedicated coverage to candidates of Portuguese descent or Portuguese-speaking candidates for the municipal elections. Furthermore, a visit by Cape Verde's president during the month of May triggered coverage by all the media here under analysis. *Contacto* (see Table 1) engages in content which covers not only the Portuguese community but that of other Portuguese-speaking countries, as it shares some of the ordeals and successes that different migrant communities share. It thus fulfils the intercultural ideal of both giving a positive image of immigration while keeping the ties between the home country and the host country. In fact, its sections are broadly divided into content/news relating to Portugal and its weekly highlights, and a local Luxembourg section where news about Luxembourg is featured.

*Contacto*'s historical evolution reveals that whereas a decade ago there was an almost complete absence, in Portuguese-language media, of coverage relating to Luxembourg politics (Cascao 2012), the situation has since evolved. These changes in editorial coverage (of Luxembourgish politics) can be interpreted as an important factor in the attempt to understand how intercultural dialogue manifests itself, but it says more about a simple keeping pace with the times changing and for a changing audience no longer composed of the "first generation" only.

In fact, in the present day, a significant amount of *Contacto* coverage relates to politics and local elections, admittedly because of the presence of numerous candidates with Portuguese descent or speaking the Portuguese language (Brazilian, Angolan, Guinean (Bissau)). Portuguese-speaking communities are, naturally, a favourite coverage topic of *Contacto*. Broadly, the representatives of the community are depicted positively or by displaying their struggles as immigrants. The newspaper also transparently covers less "colourful" news topics, as for example it did when it decided to highlight how Portuguese top the lists of domestic violence and abuse as both victims and perpetrators (Santos Ferreira 2023). The same topic, although having been highlighted by most of the other media, did not mention the nationalities behind the numbers. This widely shared editorial decision by the other minority and majority mainstream media allows for an interpretation of an intercultural principle being applied: that of showcasing immigration in a favourable light rather than highlighting its flaws.

French-Language media

In the same vein as *Contacto*, *Virgule* (see Table 1) replicated some of the stories on political runners for the municipal elections, this time portraying French-speaking candidates. Furthermore, some articles featuring immigration issues are initiated by the outlet's journalists and not always just a pure translation of the *Luxemburger Wort*.

*Virgule* focuses on content that interests cross-border workers, whilst covering the main news related to France.[31] What is striking in this publication is the neglecting of other immigrant French-speaking communities (i.e., from outside Europe) present in Luxembourg, who receive little or virtually no coverage. The issue of religious diversity illustrates the case in point, as there was no news covering the Ramadan period, for example, a factor that can be read either in light of a French republicanism hostile to the visibility of religious symbols, celebrations, and manifestations or, again, in a vision of diversity that relegates such celebrations to a private sphere in the spirit of "blending in". As for *RTL Infos*, despite displaying content that is, for the most part, a translation of the *RTL Lëtzebuerg* website, it displays some editorial independence in the shape of content such as the videos it releases promoting the lives of Portuguese in Luxembourg. It surprisingly gave much prominence to the Portuguese community, hinting again at the leverage each of the official languages may have on different minorities' media needs. It furthermore highlights the fact that the Portuguese-speaking communities are still deprived of a televised means of communication locally. The fact that RTL invests in an English-speaking audience through *RTL Today* but deprives a much more numerous Portuguese-speaking community gives a clear impression about the power dynamics of some languages against others.

## 4. Results: Mixed Narratives on Immigration and Refugees

Different media landscapes in different European countries cannot be dissociated from their countries' political visions and ways of conceiving integration and governing diversity. Europe has experienced in the last decades an increasing afflux of population movements. It has undeniably become a multicultural continent.

This research examines the practices of intercultural dialogue, arising from media production and content, in meeting the challenges brought about by immigrants and refugees in Luxembourg's increasingly diverse society. Consequently, a reflection on contemporary aspects of migration and integration cannot be dissociated from an excursion into the last few decades' backlash against the policies of multiculturalism, as well as the novel features promised by interculturalism.

At the basis of the present research's understanding of the potential role of media is the "downward" application of the concept of interculturalism at a micro level, where individuals and local institutions are delegated with the task of managing and negotiating diversity. For this reason, the relationship between immigration, media, and the local reality necessarily takes into account the role of "locality" and how it impacts aspects of integration through the focus on local media, presumably responsible for a narrative that is closer to a day-to-day experience.

Many of the challenges arising from studying migration through a prism of interculturalism are associated with the difficulty of putting the latter into practice. The media system has been identified as one potential vehicle for an intercultural type of integration.

We thus argue that that the conception of immigration that reigns in the mediated discourse of *Luxemburger Wort* and *RTL.lu* is still one where the immigrant is absent as agent, individual, or source, and deprived of visibility. That is, immigration and refugees are seldom identified but appear as abstract numbers (numbers of residents, numbers on asylum seekers, increased population) from the perspectives of the institutions, or even the nation–state. Best practices from an intercultural perspective come from a variety of minority and majority media. The Portuguese *Contacto*, French-language media *RTL Infos* and *Virgule*, and the autochthonous *De Lëtzebuerger Land* and *Tageblatt* all combine in their offered narratives content that serves both local and minority audiences as well as a balanced and overall positive narrative on immigration. Furthermore, it can be affirmed that, in the instance of the minority media examined, they successfully cross the bridge of catering for "bonding content" for their specific language community whilst covering news that affects the host society as well. The website of RTL provides the most original content in Luxembourgish with a translation in French and English. It is, however, *RTL Infos* (in French) that signals more editorial independence towards a message of interculturality. A case in point are the videos featuring the Portuguese community, which immediately gain thousands of views. In the absence of an RTL in the Portuguese language, it is understandable that Portuguese-speaking minorities turn to the French-language website in their search for representation and recognition beyond the clichés.

Journalists from the minority media owned by the Mediahuis group (see Table 1) display more editorial independence. Although they too work with content that is a pure translation of the original featured in German (*Luxemburger Wort*) into other languages, the opposite is, at times, also true. Articles run and produced by the journalists in the French and Portuguese *Vigule* and *Contacto* get, in some cases, to be translated into German and feature, at some point, in *Luxemburger Wort*. They are, however, subject to a selection process, with *Luxembourg Wort* acting as a gatekeeper to which news content is deemed of interest to convey to its autochthonous audiences. How media affects research itself has not produced a universally valid theory to date, and thus there is no clear answer to the question of the effects of mass media and its content (Weber-Menges 2015).

The role of media is not necessarily representative of the attitudes or opinions of host society. However, its role in creating and shaping language and attitudes (Foot 1999), or rather opting to not mention them, may well be more significant than that to "merely reflect them [majoritarian opinions] "(Foot 1999, p. 171). Whether by rendering invisible immigration, stories, voices, sources, actors, and faces there is an inherent intention of "normalizing" immigrants and refugees through a narrative of *blending in* remains an open question. It could hypothetically be a way for a part of the mainstream media to acknowledge their "normalisation" and full acceptance in the host society as "one of us". However, it unavoidably deprives them of a story where their diversity is rendered visible and is promoted as resource, which interculturalism, as seen, utterly preconises.

The coverage is neither negative nor problematic but rather "invisible", indeed as if immigrants' lives did not exist. Issues related to immigration (access to housing, education, the job market, housing for refugees) when and if they feature in the agenda of the main editorial groups to their autochthonous audiences are blurred under the one-size-fits-all concept of "social issues".

Conversely, the more marginal audiences of *Tageblatt* and *De Lëtzebuerger Land* (see Table 1) show more openness to stories relating to immigration, diversity, and refugees (each week each of the two newspapers runs at least one article where diversity is portrayed one way or another).

On the front of the minority media, whereas some fulfil the paradigm of a successful intercultural media integration practice—like *Contacto*, by granting balanced content between the host society, society of origin, local issues faced by the wider community (i.e., not only the Portuguese community), but also other Portuguese migrants and refugees—others fall short (*Virgule* and *Luxembourg Times*) of going beyond representing the interests of their principal audiences. This results in a visible absence of content covering English-language-speaking communities who are not British and who might not be working strictly for the financial sector in *Luxembourg Times*, and the same for *Virgule*, which focuses its content on cross-border French-speaking communities but leaves out other French-speaking communities, which are rarely and scarcely represented in its pages.

On the contrary, all minority media run content explaining the history, facts, and political highlights about Luxembourg. In this sense, the idea of a "ghettoization" promoted by some authors in relation to the risk of minority media's self-segregating practices can be dismissed. Moreover, journalists working for some of these minority media are driving the change for some stories to leave the realm of minority media only. Content created by one of *Contacto*'s journalists, who has set out for a month-long walk of Luxembourg interviewing different personalities, received translation and was followed in each of the Mediahuis outlets (see Table 1) in other languages (Rodrigues 2023).

Audience share may help explain, admittedly, the neglecting of many of the immigrant communities who are not represented in the media analysed and who might seek representation in other fora regardless of their presumable "national", "ethnic", or language belonging. This proves once again that thorough research on the appetite and trends of media consumption from different groups would be necessary.

## 5. Conclusions—Together or Side by Side?

Practices of interculturalism and multiculturalism are both at work in the mediated narratives in Luxembourg.

As seen, in qualitative terms, two of the most important print media outlets in Luxembourg (analysed) do not thrive on focusing their narrative on the stable dimension of immigration in Luxembourg or on both rendering visible and promoting immigration as a resource.

This is not the case, however, with regards to the French- and Portuguese-language media owned by the same groups (Mediahuis and RTL) and accessible through the same webpage,[32] although *Virgule* shows signs of a selected representation of difference showcasing, foremostly cross-border workers. If they are to be considered a minority media within a host society majority outlet (Mediahuis and RTL), then they do display, to an extent, the openness of host society media groups towards treating immigration under a new lens. Albeit smaller in circulation numbers/audiences, the same can be said of the *Tageblatt* and *De Lëtzebuerger Land*. This can be attributed to an editorial stance which thinks outside the traditional centre-left/centre-right boxes on offer by Luxembourgish media for more than a century. These are deeply rooted in the history of the media in the country, divided between a traditionally Catholic, conservative side and a more liberal, anti-clerical one on the other.

"Luxembourg is not grey, boring and cold", ran the editorial of the *Tageblatt* (Hamus 2023) the day after the festival of Festival des Migrations, which celebrates different immigrant communities and associations.

The paradox of Luxembourg's print media scene offer on diversity content is the following: even when it does uncontestably create a space for some communities, in particular those who are numerically superior, to express themselves, it risks creating clusters of audiences entrenched in themselves, thus contributing to a neglecting of the

narrative about those same communities to a host/majority population. By erasing their particularities and essence to make them blend in, such media therefore contribute to an assimilation of sorts, whereby backgrounds do not matter for the better or for the worse. What remains to be answered is whether that neglect is among the ultimate goals of a political interculturalism stance regarding immigration. And whether, ultimately, it answers the need for a way of reporting that leaves out people with a longstanding presence in the Luxembourgish territory and job market, therefore missing the opportunity of "normalising "their" experiences.

Consequently, a reflection on contemporary aspects of migration and integration cannot be dissociated from an excursion into the last decades' backlash against the policies of multiculturalism as well as the novelty features promised by interculturalism. At the basis of the present research's understanding of the potential role of media is the "downward" application of the concept of interculturalism at a micro level, where individuals and local institutions are delegated with the task of managing and negotiating diversity. The more visible and real side of diversity, the one that can be seen by walking out one's door, was sought in this research. It is specifically the role of "locality" and relatedness, presumably responsible for a narrative that is closer to a day-to-day experience, that is absent from the two dominant mainstream media outlets, the *RTL Lëtzebuerg* website and *Luxemburger Wort*. Within their quite limited coverage, migrants and refugees are typically represented in neutral terms when they happen to be represented, whereas in the minority media, the narrative is altogether positive, therefore reflecting an editorial line aimed at sharing the role immigrants have in the arts and culture or simply by telling their life stories. The manifestations of audiences represent a necessary endeavour for future research. The need for research on Luxembourg's online news consumption, both quantitatively and qualitatively, is something that the Media Monitoring Report also urged (Kies et al. 2022).

The high risk in terms of access to media for minorities (75%), denounced by the Media Monitoring Report, is more applicable, as indeed highlighted in the report, in the fields of radio and television rather than in the realm of print media. It is, however, crucial that minorities (and among them refugees) other than the Portuguese speakers, the French-speaking cross-border workers, and the international community with English as lingua franca also see themselves represented in the print media. Conversely, in an intercultural and meaningful interaction logic, it would be optimal that the host society too has access to content portraying positive depictions of immigrant communities and refugees that goes beyond the abstract numbers of resident populations, refugees, and who gets to vote. The voices and faces of immigration and refugees are still invisible to a great deal of the Luxembourgish print media (digital included) market, namely RTL and Mediahuis.

The interculturalist approach from other host society editorial groups is laudable, and aligned to the local and national intercultural-oriented governance agenda. Its effects and applicability in terms of some of Luxembourg's most dominant print media (those specifically aimed at host society in Luxembourgish and German) proves to be still unconvincing in the ways it translates into their master narratives and messages on immigration and refugees. There is, nevertheless, a commendable effort being made from both a part of the mainstream media and minority media in making the message of interculturalism accessible and across to their audiences.

The complex language situation of Luxembourg plays a key role in such neglect, making it so that different populations do not have a common space of encounter in the mainstream print media, thus reflecting what goes on in society at large. This is something that Luxembourgish–Tunisian director Nadia Misri explored in her short film "A place to be" and went on to clarify at one of *RTL Letzebuerg*'s most viewed programs, *Kloertext*. In it, the artist noted the crucial role language comes to play in this feeling of separateness:

> the existence of parallel societies within Luxembourg, often operating independently of each other (. . .) the absence of the Luxembourgish language automatically results in exclusion from many aspects of everyday social life. Luxembourg

is a small country, akin to a bubble, and without knowledge of the language, one finds themselves encased in a bubble within a bubble" (Mart and Rasqué 2023)

It is unrealistic to expect, at least, old Luxembourgers to make use of minority media. For this reason, an appropriate representation of the country's minorities is urged in some of Luxembourg's dominant mainstream media.

In a country where immigration stories remain largely untold in some of the most consumed autochthonous media, by choosing not to render them visible in their individuality, the risk is that of entering the predicament repeatedly criticised about multiculturalism.

The result, therefore, is that on many fronts, the Luxembourgish society is paralysed at an interculturalism of theory only. The dominant practice remains, in the two main mainstream media outlets of the host society, RTL.lu and Wort, paradoxically, a translating of the aspects accused of having been overlooked by multiculturalism, thus creating "parallel societies" whose paths never positively meet or interact. The result is a weakening of the theoretical dialogue on interculturalism mainstreamed at a political level but out of step with its concrete and pragmatic applications in the field of media. Further research should be directed at achieving a better understanding of how audiences perceive media content framing diversity, as well as at the challenges multilingualism poses in terms of communities accessing information about one another.

**Supplementary Materials:** The following supporting information can be downloaded at: https://www.mdpi.com/article/10.3390/socsci12110589/s1: Excel spreadsheet: content extract 2023.

**Funding:** This research received no external funding.

**Institutional Review Board Statement:** Not applicable.

**Informed Consent Statement:** Informed consent was obtained from all subjects involved in the study.

**Data Availability Statement:** The data presented in this study are available in the supplementary material above.

**Conflicts of Interest:** The author declares no conflict of interest.

## Notes

1  The old Integration Act (*Integrationsgesetz*) of 2008 is to be replaced by the new Act on Intercultural Coexistence—a text that should mark the culmination of almost ten years of work in the Ministry of Integration for Minister of Integration Corinne Cahen (DP).

2  Some of the most informed contemporary debates and studies on multiculturalism and interculturalism cannot be viewed separately from the different perspectives that Canadian scholarship has to offer on the matter. For a thorough understanding, see (Kymlicka 2015; Taylor 2012; Bouchard and Taylor 2008).

3  For the sake of clarity, we will be referring to the host society as the majority and foreign residents as minorities.

4  If we remove the Luxembourgers from the population equation and consider only foreigners, the weight of the Portuguese community (30.8%) is significant. In Luxembourg, one in every three foreign residents is of Portuguese nationality, and there are municipalities where the percentage of Portuguese among foreigners reaches 70%.

5  Very rooted in historical tradition, despite Luxembourgish being an official language since 1984, literacy in public schools is still held in German. With the support of the current Ministry of Education, a French-language literacy curriculum has been launched with a pilot project in some schools in south of Luxembourg as well as with the introduction of a few public international schools scattered (where pupils can choose their main literacy language between German, French, and English) around the country. Despite an increased acknowledgment of the need for a change, progress in making the offer widespread is slow. The use of German, besides its literacy function, is relegated to that of the national print media. Overall, the German language has been losing ground to French language in the past few decades (Fehlen 2013) and is progressively being abandoned in official administrative communications with the public. The decay in the use of the German language along with its effect on the school results of children from migratory backgrounds has, after decades, been acknowledged by the Ministry of Education, which is slowly adapting the education system to allow an at least parallel offer of education curricula in French. For more on the history of the evolution of the German language in Luxembourg, see (Sieburg 2013).

6  "Allowing pupils to be taught in one main language of their choice could greatly improve results, European Commission report recommends", the Luxembourg Times reports (Velasquez 2023).

7    Exponents from both the Democratic Party (DP) and the Socialist Party (LSAP) advocate for a wider offer of public schools offering a curriculum in French to meet the needs of an ever-growing immigrant population.

8    "Trilingualism is a phenomenon of great concern to Luxembourgers. Most of them are very attached to it, but wonder what consequences it has on their psyche (. . .) Some even express themselves remarkably in French or German, but rarely in both (. . .) Plurilingualism has been blamed for a certain sterility in literature and a lack of creative inspiration". (Trausch 2007).

9    The more conservative population expresses concerns about the widening of the offer of a literacy path in French, whereas the *new* Luxembourgers, and among them those of migratory background, are in favour of a literacy path in the language the child is more familiar with. The latter has been namely supported by significant academic research as well as gathering political support from the center and the left political parties.

10   Although having full political recognition, Luxembourgish struggles to assert and "emancipate" itself from German as a support literacy language.

11   A note on terminology: Like all other language choices on this topic, (*community*, *ethnic*, or *diasporic*,) the term minority is imperfect, but nevertheless the most appropriate to represent the realities that we here want to differentiate.

12   Intercultural media integration builds upon the core elements of interculturalism, of which the active acceptance of the necessity of immigration and the advocacy for dual citizenship are perceived as key elements in a successful integration process. With regard to access to nationality, Luxembourg has shown a "Copernican revolution " (Scuto 2023) by facilitating access to nationality in the past few decades.

13   Its audience being traditionally conservative and Catholic, historically close to the centre right wing party CSV.

14   *Tageblatt* is the second largest newspaper in Luxembourg, behind *Luxemburger Wort*. At its inception, the newspaper was directed by one of Luxembourg's finest intellectuals of the interwar period, Frantz Clément. The *Escher Tageblatt*, being located at the heart of the southern industrial region, started out as a socialist newspaper and has ever since kept a left-wing blueprint to it.

15   "Since it was founded in 1954, by Carlo Hemmer, *Lëtzebuerger Land* was intended to fill the void created by the disappearance of the pre-war liberal press and to provide a forum for writers who did not want to express themselves in the partisan dailies. (. . .) From the outset, the paper tried to differentiate itself from the daily press through its detailed analytical articles and high-level discussion forums. Although *the Lëtzebuerger Land* was initially suspected of being an organ of big industry, it has always opened its pages to social and ecological issues." (Extract from the newspaper's history webpage, available at https://www.land.lu/online/www/menu_content/history/FRE/index.html (accessed on 13 July 2023).

16   Source: (TNS ILRES 2020).

17   For instance, *L'essentiel*, a free daily belonging to the Editpress group, and *le Quotidien*, a French-language newspaper with a circulation of 21,200.

18   All three newspapers will typically include some articles, columns or opinions in the French language too, in their weekly print and digital editions, in the case of *Luxemburger Wort* to a lesser extent.

19   *RTL Lëtzebuerg* displays content in Luxembourgish, whereas its "sister" webpages *RTL Infos* and *RTL Today* display content in French and English, respectively.

20   Digital accesses excluded.

21   Not in circulation at the time of the survey TNS.

22   Circulation of 7500 in 2022 of its monthly magazine versions. Digital audience numbers not available.

23   see Note 16 above.

24   In Luxembourg, there is one independent community media, *Radio ARA*, which is state-funded with an outreach to different communities in multiple languages: English, Spanish, Arabic, and Italian.

25   Both English-language weeklies *Paperjam* and *Delano* address the international community; their editorial line focuses on business life and national issues with "interviews with expats and presents local decision-makers" (according to its editorial line).

26   We hereby, naturally, exclude second- and third-generation Luxembourgish–Portuguese. Included are, thus, the first generation of Portuguese and a new wave of Portuguese who have been steadily reaching Luxembourg for the past 15/20 years and for whom their first foreign language is English. For more on the nuances of the different clusters of Portuguese immigration, see Aline Schiltz (Schiltz 2018) and the work of Heidi Martins and Anne Carolina Ramos (Ramos and Martins 2020).

27   Accessing content on the RTL platforms necessarily included video content, as they are an integral part of the digital products developed by RTL in the last few years.

28   With the exceptions of the weekends of 1–2 April and 27–28 May for the paper versions of *Wort* and *Tageblatt*.

29   In the original: Glauben Sie im Ernst, dass Luxemburger sich am Wochenende mit nicht Luxemburgern in einen Kurs setzen werden, in dem ihnen das politische System des Landes erklärt wird?

30   Total of 2268 requests for international protection in 2022. Source: Ministère des Affaires Etrangères et européennes, Direction de l'immigration.

31   From *Virgule*'s website: "A new logo and a new identity, but the same concern: to offer quality journalism to French-speaking readers living in France and cross-border workers."

32    Unsurprisingly, subscriptions for each of the media outlets owned by Mediahuis need to be made separately, and there is no overall subscription covering all four: *Luxembourger Wort*, *Contacto*, *Virgule*, and *Luxembourg Times*.

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
