# Peer review of "Between an Acknowledgment of Immigration and Neglect? Assessing Interculturalism and Media Integration in Luxembourg"

_socsci, doi:10.3390/socsci12110589_

Round 1

Reviewer 1 Report

The authors are interested in examining whether and how mainstream and minority media outlets promote integration through intercultural dialogue. While the topic is an interesting and important one, I have several concerns. 

First and foremost, the manuscript was somewhat difficult to follow. At several points throughout my reading I had to stop and re-read sentences or paragraphs multiple times to be able to follow the authors' train of thought. Some of this may be due to English language issues, but many sentences felt written as a stream-of-consciousness, rather than structured academic writing. I recommend a thorough editing of the manuscript for readability and clarity. 

Second, the authors write about multiculturalism and interculturism without ever clearly defining them, particularly for the current study. I recommend doing so early on as a way to help readers digest the material.

Relatedly, when discussing their method, the authors state that they use "the model of intercultural media integration model as blueprint," (p. 6), but they should explain exactly what they mean by this. More clarity is needed about their methodological process. 

Further, they state on page 9 that they are not "concerned with finding the totality of occurrences featured in each of the newspapers, but rather reflect on the occasions such topics did or did not receive coverage." However, it is unclear how they can determine the occasions such topics did not receive coverage if they are not reviewing the totality of occurrences.

As stated in my comments to the authors, I am unsure whether my difficulty following the authors train of thought, logic, and reasoning throughout the manuscript was due to writing quality or difficulties with the English language. 

Reviewer 2 Report

This study examines the representation of immigrants and refugees in Luxembourg media, the paper strongly argues that although immigrants do have a voice it is only in specific media outlets, but not those that are considered mainstream. The study is well written, analyzed, and well argued. 

The paper was clear, and addressed an important issue related to immigrant voices. It thoroughly covered the internal and border issues of the area, and connected it to the literature.

The explanation of the methods used is lacking. What approach was taken to coding the data, was it thematic coding? What were these codes? As this appears to be a qualitative analysis how as saturation reached? Beyond this, the authors briefly touched on the importance of the findings at the end, but need to really expand on the implications of the study (theoretical, program, policy) to really show the significance of the project.

Reviewer 3 Report

The article approach a very interesting topic about the "visibility" (or not) of immigrants and refugees in the national and local media and, from this analysis, assesses the practical application of formally instituted intercultural integration policies, many of which are directly linked to the country's status as a Member State of the European Union, which establishes common policies.

This is a country in which around 50 per cent of the resident population is of foreign origin, yet this population is not fully integrated into Luxembourg society.

Considering that this is a very relevant topic, I think it's worth reflecting on and analysing certain aspects, firstly, why does the failure to mention immigrants and refugees in the media constitute a sociological problem? From my perspective, this problem should be explored in greater depth, and it involves the power relations between the "established" and the "outsiders" (Elias & Scotson, 1994, "The Established and the Outsiders", SAGE PUBLICATIONS LTD).

https://uk.sagepub.com/en-gb/eur/the-established-and-the-outsiders/book203270

But also the perspectives on the "foreigner", the "stranger" (Simmel, Georg), always seen as a threat to social stability. In this case, it is important to analyse whether we are dealing with some kind of "whitewashing", albeit encapsulated, of the phenomenon of immigration and immigrants, distinguishing who are good immigrants and who are not, and this often depends on the country of origin and its "submission" and "docility". Therefore, this condition will define who deserves to be welcomed and who doesn't. For this approach, it is important to adopt a non-dichotomous perspective, one that is not based on "us" and "them" but on the process of building social relations and their transformations, which are not crystallised over time. It would be interesting to bring this more fluid approach to the text, only in this way will it be possible to penetrate the intricacies of understanding the process of acculturation and adaptation between those who arrive and those who stay. It's never a unilateral or static process!

Another concept that I think I can bring to the discussion is power relations: obviously this relationship is always underlying between the "natives", the "established" and those who arrive, the "outsiders", the foreigners. Even in this country where there is such a large number of long-standing immigrants (several generations), this distinction remains. In this regard, I recommend reading Bourdieu, P. (1979). La Distinction: Critique Sociale du Jugement Paris: Les Éditions de Minuit and Pierre Bourdieu, "Sur le pouvoir symbolique", Annales. Histoire, Sciences Sociales, vol. 32, no 3, 1977, p. 405-411 https://www.persee.fr/docAsPDF/ahess_0395-2649_1977_num_32_3_293828.pdf This question emerges when referring to the mastery of some of the official languages less used in everyday life, which are used by the Luxembourg elites and also by the main media.

I also think the concept of identity and its process is important for framing the complexity: what does it mean to be Luxembourgish today? (please look up “Le pouvoir de l'identité”, Manuel Castells) and Amin Maalouf, « Les Identités meurtrières ».

There are a few points that I think should be clarified:

- The author sometimes refers to immigrants, sometimes to immigrants and refugees. Given that the legal and formal statuses are different, I suggest that the focus should only be on immigrants, so much so that no theoretical framework is developed for refugees in the text (which is different from immigrants);

- Regarding the concepts of multiculturalism, interculturalism and intercultural dialogue, their presentation tends to be too abstract. I suggest complementing this presentation with examples from everyday life - social relations in everyday life often deconstruct the conceptualisation of phenomena;

- We also suggest distinguishing between the concept of integration in terms of official policies and social integration in a sociological sense. As mentioned by SCHNAPPER, Dominique (Qu'est-ce que l'intégration? Paris: Éditions Gallimard: folio actuel Inédit, 2007) there are dimensions to integration and exclusion: nobody is totally excluded.  People are not "islands", they are not isolated from others, they work, they have children, they attend public and private institutions, etc.

As for the methodology, I suggest identifying the categories and subcategories used and the criteria used to define them. This information is a bit diffuse. It would also be useful to explain more about how the content analysis was carried out.

In order to better distinguish the sections of the article, it would be useful to create a Results and Conclusions section, which could expediently summarise the results and the article's contributions to the advancement of knowledge.
